# Validation of the Liver Disease Quality of Life Instrument 1.0 in Patients with Chronic Hepatitis B: A Prospective Study

**DOI:** 10.3390/jcm8050656

**Published:** 2019-05-10

**Authors:** Yeonjung Ha, Sohyun Hwang, Young Eun Chon, Mi Na Kim, Joo Ho Lee, Seong Gyu Hwang

**Affiliations:** 1Department of Gastroenterology, CHA Bundang Medical Center, CHA University; 59 Yatap-ro, Bundang-gu, Seongnam-si, Gyeonggi-do 13496, Korea; nachivysoo@chamc.co.kr (Y.E.C.); mina2015@cha.ac.kr (M.N.K.); ljh0505@cha.ac.kr (J.H.L); sghwang@cha.ac.kr (S.G.H.); 2Department of Pathology, CHA Bundang Medical Center, CHA University; 59 Yatap-ro, Bundang-gu, Seongnam-si, Gyeonggi-do 13496, Korea; blissfulwin@chamc.co.kr

**Keywords:** Quality of life, Liver Disease Quality of Life Instrument 1.0, Validation, Chronic hepatitis B

## Abstract

The purpose of this study was to report on the clinical usefulness of the Liver Disease Quality of Life Instrument (LDQOL) 1.0, which was prospectively measured in chronic hepatitis B patients. We regularly followed up with patients with chronic hepatitis B between 2008 and 2010 who were enrolled in the study, and the LDQOL 1.0 was filled out until 2015. The reliability and construct validity were evaluated by Cronbach’s α values and analysis of variance. Cox proportional hazards models were used to identify questionnaire components associated with death and decompensation. The LDQOL 1.0 scores were compared between groups of patients with different clinical characteristics. A total of 192 patients (27.1% with cirrhosis) were enrolled. The LDQOL 1.0 was reliable with high internal consistency based on the Cronbach’s α value. Most of each component was significantly associated with liver disease-related parameters, such as disability days, self-rated severity of liver disease symptoms, and Child-Pugh class. The change in concentration score between the first and last visit significantly predicted death (hazard ratio (HR), 0.44) and decompensation (HR, 0.97; *p* < 0.05 for both). Patients who achieved complete viral suppression did not show better scores than those who did not. In conclusion, the LDQOL 1.0 was prospectively validated in patients with chronic hepatitis B. Complete viral suppression did not influence the improvement of quality of life scores. The change in concentration scores over time was predictive of death and decompensation.

## 1. Introduction

Patients with chronic health problems have impaired health-related quality of life (QOL) [1,2]. The usual tests conducted in the clinic generally focus on the objective aspect; however, these tests do not necessarily correlate with the subjective perception of health [1,3]. Because patients’ QOL is associated with clinical outcomes, it is of great importance to evaluate QOL relevantly [4].

The tools for measuring QOL can be categorized into generic or disease-specific instruments [1]. A hybrid tool incorporating those two cores, the Liver Disease Quality of Life Instrument (LDQOL) 1.0, was developed and validated in 2001 [5,6]. Our group translated the LDQOL 1.0 into Korean and showed its usefulness in patients with chronic liver disease [7]. However, only the liver disease-specific core was translated and validated at the time, and the questionnaire was completed only once by patients with liver disease of various etiologies. 

Chronic hepatitis B (CHB) is the most common cause of chronic liver disease in the Asia-Pacific area, including South Korea, and the clinical outcomes are improving thanks to the nucleos(t)ide analogs (NUCs) treatment [8]. However, it is not clear whether the QOL is also improved following such treatments.

Therefore, in this study, we tried to validate the complete version of the LDQOL 1.0 incorporating both the generic and liver disease-specific core in a prospective cohort of CHB patients. Additionally, we checked the changes in scores over time and compared them between a subgroup of patients with different viral replication statuses. Finally, we identified the LDQOL 1.0 components associated with clinical outcomes, i.e., overall survival (OS) and decompensation.

## 2. Materials and Methods

### 2.1. Study Design

This prospective study was done in a single center in South Korea. Eligible patients were ≥ 18 years old and came in for regular follow-ups for more than three months at the outpatient clinic for CHB. Screening and enrollment were done between 2008 and 2010, and the LDQOL 1.0 was filled out up until 2015. Baseline characteristics and follow-up and mortality data were fully accessible through the medical record.

The study was conducted in accordance with the ethics principles of the Declaration of Helsinki and International Council for Harmonization Good Clinical Practice Guidelines. The final protocol and informed consent forms were approved by the institutional review board (IRB No. 2008-064) and all patients provided written informed consent.

### 2.2. Liver Disease Quality of Life Instrument 1.0 Measurement

The Liver Disease Quality Of Life Instrument (LDQOL)1.0 consists of generic and liver disease-specific cores. The generic core utilizes the Short Form-36 (SF-36; version 2.0) that includes eight components and 35 items. One single item that asks about changes in the respondent’s general health compared to last year is also included (a total of 36 items). The liver disease-specific core has 12 components comprising 75 items. The LDQOL 1.0 was repeatedly completed and the interval was determined according to the physician’s discretion. A blinded researcher reviewed the completed questionnaires and transformed the original score to a 100 scale by converting the worst answer to zero and the best answer to 100.

### 2.3. Statistical Analysis 

Cronbach’s α values were calculated to evaluate the internal consistency. The correlation between each component was presented as Pearson’s r. For assessing construct validity, an analysis of variance (ANOVA) was performed to evaluate the association between each questionnaire component and the common parameters of liver disease. The common parameters of liver disease included the following four items: (1) self-reported disability days (0 vs. 1–10 vs. 11–20 vs. ≥ 21 days in the preceding month), (2) self-rated severity of liver disease symptoms (from 1 (no symptoms) to 5 (extremely severe symptoms)), (3) Child-Pugh class (A vs. B vs. C), and (4) duration of liver disease (from 1 (less than 6 months) to 7 (more than 10 years) based on the diagnosis date on the medical record). The results of ANOVA were presented with the *F* values showing differences (*F* = $\frac{\mathrm{variance}\;\mathrm{between}\;\mathrm{groups}}{\mathrm{variance}\;\mathrm{within}\;\mathrm{groups}}$) and the *p*-values were adjusted by the Benjamin–Hochberg method. 

According to the median number of LDQOL 1.0 completions, the scores were grouped into first and last scores. The scores completed in the same calendar year were averaged together. If all of the scores were from different calendar years, values that showed a higher approximation in absolute numbers were grouped together (e.g., 10 points in 2007, 15 points in 2008, 70 points in 2009, and 30 points in 2010 were grouped as 12.5 points (average of 10 and 15 points; first score) and 50 points (average of 70 and 30 points; last score).

The Cox proportional hazards model was used to evaluate the LDQOL 1.0 components predictive of OS and decompensation. The OS duration was calculated by subtracting the date of the first questionnaire from the date of death. Decompensation was defined as the first clinical evidence of ascites, jaundice, variceal bleeding, or hepatic encephalopathy in patients who had not had these conditions before. Patients were followed up with until death or Dec 31st, 2016. In case of a follow-up loss, data were censored at the date of the last clinic visit.

Finally, subgroup comparison between patients with or without complete viral suppression was performed. The viral suppression status was evaluated by real-time PCR assay (Roche Diagnostics) when patients completed the last LDQOL 1.0 questionnaire and defined as undetectable hepatitis B virus (HBV) DNA in the serum. 

All reported *p*-values were two-sided, and the *p*-values < 0.05 were considered statistically significant. Statistical analyses were conducted using the SPSS package (SPSS version 22.0 for Windows; SPSS Inc., Chicago, IL, USA).

## 3. Results

### 3.1. Baseline Characteristics

A total of 192 enrolled patients completed the LDQOL 1.0 multiple times (median, 2; interquartile range, 2‒4 times). Baseline characteristics are summarized in Table 1. The mean age was 41.3 years, 72.4% were male, and 27.1% had liver cirrhosis. Approximately one fifth of patients were not treated with NUCs, and adefovir was the most commonly prescribed drug (30.5%) among patients receiving NUCs. The median HBV DNA levels were 1.45 × 10^5^ copies/mL.

### 3.2. Descriptive Statistics

The concentration score was the highest (mean, 91.6; standard deviation (SD), 14.9), and the general health perceptions score was the lowest (55.0 ± 19.4) (Table 2). The physical components showed a higher score (physical functioning: 88.5 ± 17.9; bodily pain: 88.5 ± 17.1) compared to the emotional component (emotional well-being: 71.4 ± 18.4) in the SF-36. The average scores of cognitive components were higher (concentration: 91.6 ± 14.9; memory: 90.4 ± 12.5) than those of emotional components (hopelessness: 68.8 ± 20.8; health distress: 74.7 ± 23.1) in the liver disease-specific core. The internal consistency was high with the Cronbach’s α values > 0.70 in all components except for quality of social interaction (0.65) and sleep (0.69).

The correlation of each component is presented in Table 3. Within the generic core, bodily pain and emotional well-being showed the weakest correlation (Pearson’s r = 0.32), while role limitations due to physical and emotional health problems were strongly correlated (r = 0.85). Sexual functioning and sexual problems showed a close correlation (r = 0.74); on the other hand, memory and stigma of liver disease were weakly correlated (r = 0.26). The strongest inter-correlation was found between role limitations due to physical health problems and concentration (r = 0.74). Emotional well-being and sexual functioning were shown to have the weakest correlation (r = 0.18).

### 3.3. Construct Validity

The *F* values in Table 4 represent the differences in each component score according to the different severity of liver disease. All the component scores significantly differed according to the self-reported disability days and self-rated severity of liver disease symptoms. The ‘role limitations due to physical health problems’ component indicated large differences in scores between patients who rated their liver disease symptoms as more vs. less severe (*F* = 29.24; *p* < 0.001). The scores also significantly differed according to Child-Pugh class for 13 out of 20 components. On the other hand, all of the components showed comparable scores, regardless of the duration of patients’ liver disease (all *p* = 0.94). 

### 3.4. Clinical Outcomes

As we focused on the change in scores over time, the scores of each component were grouped and averaged together according to the rules described in the Materials and Methods section, and the difference between the first and last score was calculated and included in the analysis. 

On the univariate analysis, a change in concentration scores significantly predicted OS (per 10 point in 100 scale; hazard ratio (HR), 0.44; 95% confidence interval (CI), 0.24–0.78; *p* = 0.005) and decompensation (per 10 point; HR, 0.97; 95% CI, 0.94–1.00; *p* = 0.029) (Table 5). Multivariate analysis was not conducted because of a small number of events occurred (5 dead; 14 decompensated).

### 3.5. Subgroup Comparison

To compare the change in the LDQOL 1.0 scores between patients with different clinical characteristics, subgroup analysis was conducted. 

Patients who achieved complete viral suppression on their last visit were compared with those who did not (Table 6). The scores of all components did not show significant difference between the two groups at last visit. The general health perceptions score was increased by an average of 5.69 points in the 100 scale (SD, 20.18) in the suppression group compared to an average of 0.61 point decrease (SD, 14.61) in the non-suppression group (*p* = 0.037). On the other hand, the memory score significantly decreased in the suppression group (mean ± SD, −5.88 ± 12.13) compared to the non-suppression group (mean ± SD, +0.57 ± 15.43; *p* = 0.006). The change in other scores was comparable.

## 4. Discussion

The complete version of the LDQOL 1.0 was first validated in CHB patients in South Korea. The LDQOL 1.0 was reliable and valid, being highly associated with the common parameters of liver disease. The change in concentration score over time significantly predicted OS and decompensation. Patients achieving complete viral suppression did not necessarily show greater improvement of scores, suggesting the need for regular QOL assessment regardless of viral replication status.

The goal of CHB treatment is not only to reduce death or hepatocellular carcinoma but also to improve QOL [8,9]. NUCs treatment significantly reduced liver inflammation, the progression to cirrhosis, and the development of hepatocellular carcinoma in CHB patients [8,9]. However, the objective health status does not necessarily reflect patients’ physical or emotional capacity [1,3], while self-perceived health status is an important predictor of death [4].

A previous study that measured the general aspects of QOL demonstrated HBV infection itself or NUCs treatment was not associated with the change in QOL [10]. On the other hand, a recent study by Younossi et al. reported that the decrement of HBV DNA improved some QOL scores; however, complete viral suppression was not associated with the improvement of QOL [11]. Another study that included a large number of NUCs-treated patients found that QOL scores were improved in patients who had a virologic response [12]. However, these studies only included patients receiving NUCs and used relatively simpler instruments compared to the LDQOL 1.0. Additionally, Younossi et al. analyzed the data from clinical trials, which might not reflect a real clinical setting [11].

Therefore, we enrolled a more diverse patient population, including those who did not receive NUCs, and used the LDQOL 1.0 as a more comprehensive and detailed tool, and the patients filled out the questionnaire repeatedly over time. By doing so, we were able to validate the LDQOL 1.0 in the types of patients that physicians encounter every day in a real clinical practice.

A vast majority of components under the LDQOL 1.0 were significantly associated with the parameters of liver disease, such as self-reported disability days, self-rated severity of liver disease symptoms, and Child-Pugh class. On the other hand, the duration of liver disease was not correlated with any of the components, in line with a previous study [5]. The reason for this may be that the duration of liver disease was estimated by the authors based on the diagnosis date on the medical record. However, most CHB patients in South Korea are infected through vertical transmission [8], therefore, the actual duration of disease should be longer.

Meanwhile, the change in concentration score significantly predicted death and decompensation. The risk of death and decompensation decreased by 56% and 3%, respectively, when the concentration score increased by 10 points on a 100 scale. In a previous QOL study involving lymphoma patients, concentration and memory difficulties were associated with anxiety, depression, or fatigue [13]. Additionally, some studies found that the QOL score of the physical components worsened only in the later stages of the disease in CHB patients [14,15], suggesting a special need for mental health evaluation in those patients. Nonetheless, the situation could be too sensitive to directly mention mental issues, or there may not be enough time for patients to complete the whole questionnaire at the outpatient clinic. Under such circumstances, physicians can briefly ask about concentration changes. This simple evaluation would enable physicians to assess patients more comprehensively. 

One caveat is that the LDQOL 1.0 was initially developed as a discriminative instrument. The discriminative instrument demonstrates the difference between patients at a specific time point [1]. The other kind of tool is an evaluative instrument, which assesses changes of QOL over time [1]. We utilized the LDQOL 1.0 with an evaluative purpose, therefore, it may be a limitation of our study. However, it is more economic to utilize the existing tool than to develop a new one. Furthermore, our study validated the LDQOL 1.0 as a discriminative tool first and then demonstrated its potential usefulness as an evaluative tool.

It is also important to note that adefovir was the most common NUC in this study. Currently, more potent NUCs that have a higher genetic barrier to resistance are used as a first-line therapy [8,9]. Because this study was initiated in 2008, our cohort might not represent current patients. However, we believe our findings can be applied to the current clinical practice, based on the results derived by subgroup comparisons.

## 5. Conclusions

We validated the complete Korean version of the LDQOL 1.0 in a prospective cohort of CHB patients with diverse clinical characteristics. Patients with favorable clinical features over time (achieving complete viral suppression) did not necessarily show a greater extent of improvement in QOL. The decrease in concentration score predicted death and decompensation. Regular assessment of the LDQOL 1.0 would be helpful and might be utilized further in developing clinical or policy guidelines. 

## Figures and Tables

**Table 1 jcm-08-00656-t001:** Main baseline characteristics of the 192 patients who responded to the questionnaire. *

Variables	All (*N* = 192)
**Mean age, years**	41.3 ± 10.0
**Male sex, n (%)**	139 (72.4)
**Marital status, n (%)**	
Single	29 (15.1)
Married	148 (77.1)
Separated	1 (0.5)
Divorced	3 (1.6)
Widowed	7 (3.6)
No response	4 (2.1)
**Education level, n (%)**	
Ninth grade or less	20 (10.4)
Some high school	68 (35.4)
College degree	27 (14.1)
University degree	66 (34.4)
Professional or graduate degree	7 (3.6)
No response	4 (2.1)
**Employment status, n (%)**	
Working full-time	121 (63.0)
Working part-time	17 (8.8)
Unemployed	1 (0.5)
Retired	3 (1.6)
Disabled	9 (4.7)
In school	12 (6.3)
Homemaker	23 (12.0)
None of the above	1 (0.5)
No response	5 (2.6)
**Health insurance coverage, n (%)**	
National health insurance program	119 (62.0)
Medical aid	5 (2.6)
None	34 (17.7)
Not sure	24 (12.5)
No response	10 (5.2)
**Total household income, n (%)**	
<$5,000	18 (9.4)
$5,001‒$10,000	17 (8.8)
$10,001‒$25,000	20 (10.4)
$25,001‒$50,000	57 (29.7)
$50,001‒$75,000	35 (18.3)
>$75,000	17 (8.8)
Not sure	18 (9.4)
No response	10 (5.2)
**Comorbidities, n (%)**	
Hypertension	13 (6.8)
Diabetes	9 (4.7)
Cardiovascular disease other than hypertension	2 (1.0)
Kidney disease	1 (0.5)
Psychiatric disease	1 (0.5)
Cirrhosis, n (%)	52 (27.1)
History of decompensation, n (%)	14 (7.3)
History of hepatocellular carcinoma, n (%)	2 (1.0)
**Child-Pugh class, n (%)**	
A	178 (92.7)
B	8 (4.2)
C	6 (3.1)
**Antiviral agent, n (%)**	
None	38 (19.8)
Adefovir	47 (24.5)
Lamivudine	33 (17.2)
Telbivudine	27 (14.0)
Entecavir	24 (12.5)
Other†	23 (12.0)
**Hepatitis B virus DNA, copies/mL**	1.45 × 10^5^ (undetectable, 1.66 × 10^7^)
**Creatinine, mg/dL**	1.0 ± 0.2
**Albumin, g/dL**	4.2 ± 0.7
**Aspartate aminotransferase, IU/L**	53.5 ± 44.6
**Alanine aminotransferase, IU/L**	64.4 ± 61.2
**Bilirubin, mg/dL**	1.0 ± 0.7
**Prothrombin time, international normalized ratio**	1.1 ± 0.3
**Hemoglobin, g/dL**	14.6 ± 1.8
**Platelet, x10^3^/μL**	168.4 ± 69.7

* Data are presented with mean ± standard deviations or numbers with a percentage or median with interquartile ranges. † Including seven patients treated with clevudine, four patients treated with adefovir and lamivudine, three patients treated with adefovir and entecavir, and nine patients with investigational drugs.

**Table 2 jcm-08-00656-t002:** Baseline descriptive statistics and internal consistency of the Korean LDQOL 1.0.

LDQOL 1.0	Number of Items	Mean Score	Standard Deviation	Minimum Score	Maximum Score	% Scoring the Floor	% Scoring the Ceiling	Cronbach’s α
**Short Form-36 core**								
**Physical functioning**	10	88.5	17.9	0.0	100.0	0.5	39.1	0.91
**Role limitations - physical**	4	83.6	23.9	0.0	100.0	1.6	46.9	0.91
**Role limitations - emotional**	3	82.7	24.8	0.0	100.0	1.0	50.5	0.91
**Social functioning**	2	85.0	19.9	12.5	100.0	0.0	49.5	0.75
**Bodily pain**	2	88.5	17.1	22.5	100.0	0.0	53.6	0.83
**Energy/fatigue**	4	58.2	21.9	6.25	100.0	0.0	2.1	0.78
**Emotional well-being**	5	71.4	18.4	20.0	100.0	0.0	3.6	0.77
**General health perceptions**	5	55.0	19.4	5.0	100.0	0.0	0.5	0.78
**Liver disease-specific core**								
**Symptoms of liver disease**	17	86.2	13.5	8.2	100.0	0.0	9.4	0.85
**Effects of liver disease**	10	82.4	12.9	36.0	100.0	0.0	9.4	0.85
**Concentration**	7	91.6	14.9	25.0	100.0	0.0	55.5	0.93
**Memory**	6	90.4	12.5	25.0	100.0	0.0	43.2	0.88
**Quality of social interaction**	5	75.1	14.9	20.0	100.0	0.0	2.1	0.65
**Health distress**	4	74.7	23.1	0.0	100.0	1.6	19.3	0.94
**Sleep**	5	63.9	15.9	20.0	100.0	0.0	1.0	0.69
**Loneliness**	5	80.8	16.9	5.0	100.0	0.0	10.9	0.73
**Hopelessness**	4	68.8	20.8	0.0	100.0	1.0	12.5	0.76
**Stigmata of liver disease**	6	80.6	21.7	0.0	100.0	0.5	30.7	0.89
**Sexual functioning**	3	89.2	16.9	25.0	100.0	0.0	39.1	0.87
**Sexual problems**	3	82.1	19.2	0.0	100.0	0.7	18.1	0.85

LDQOL, liver disease quality of life instrument.

**Table 3 jcm-08-00656-t003:** LDQOL 1.0 product-moment correlation coefficients.

LDQOL 1.0		Short Form-36 Core								Liver Disease-Specific Core											
		PF	RLp	RLe	SF	BP	EF	EWB	GH	SxLD	EfLD	Conc	Mem	QSI	HD	SL	Lone	Hope	StLD	SexF	SexP
Short Form-36 core	PF	1																			
	RLp	0.71	1																		
	RLe	0.69	0.85	1																	
	SF	0.47	0.71	0.66	1																
	BP	0.44	0.52	0.45	0.51	1															
	EF	0.42	0.56	0.48	0.50	0.37	1														
	EWB	0.37	0.45	0.45	0.48	0.32	0.69	1													
	GH	0.40	0.53	0.38	0.40	0.39	0.61	0.51	1												
Liver disease-targeted core	SxLD	0.68	0.72	0.65	0.59	0.61	0.52	0.40	0.55	1											
	EfLD	0.48	0.67	0.57	0.56	0.49	0.39	0.34	0.50	0.62	1										
	Conc	0.64	0.74	0.73	0.66	0.48	0.48	0.41	0.45	0.66	0.58	1									
	Mem	0.46	0.36	0.44	0.31	0.25	0.22	0.19	0.24	0.43	0.30	0.63	1								
	QSI	0.35	0.46	0.43	0.34	0.30	0.38	0.45	0.36	0.42	0.27	0.52	0.37	1							
	HD	0.38	0.51	0.39	0.44	0.38	0.59	0.54	0.66	0.47	0.51	0.53	0.31	0.32	1						
	SL	0.38	0.38	0.38	0.36	0.29	0.45	0.36	0.43	0.43	0.35	0.49	0.29	0.31	0.43	1					
	Lone	0.47	0.46	0.55	0.43	0.25	0.43	0.51	0.37	0.32	0.34	0.62	0.46	0.48	0.43	0.39	1				
	Hope	0.38	0.38	0.36	0.34	0.25	0.49	0.49	0.53	0.38	0.36	0.53	0.39	0.43	0.51	0.46	0.49	1			
	StLD	0.45	0.47	0.44	0.42	0.36	0.39	0.40	0.50	0.41	0.43	0.54	0.26	0.39	0.57	0.38	0.48	0.49	1		
	SexF	0.31	0.37	0.41	0.28	0.23	0.29	0.18	0.37	0.31	0.35	0.49	0.45	0.33	0.37	0.43	0.46	0.44	0.51	1	
	SexP	0.31	0.47	0.47	0.34	0.32	0.38	0.32	0.42	0.51	0.48	0.57	0.41	0.30	0.47	0.40	0.43	0.44	0.49	0.74	1

LDQOL, liver disease quality of life instrument; PF, physical functioning; RLp, role limitations due to physical health problems; RLe, role limitations due to emotional health problems; SF, social functioning; BP, bodily pain; EF, energy/fatigue; EWB, emotional well-being; GH, general health perceptions; SxLD, symptoms of liver disease; EfLD, effects of liver disease on activities of daily living; Conc, concentration; Mem, memory; QSI, quality of social interaction; HD, health distress; SL, sleep; Lone, loneliness; Hope, hopelessness; StLD, stigma of liver disease; SexF, sexual functioning; SexP, sexual problems.

**Table 4 jcm-08-00656-t004:** Construct validity of the Korean LDQOL 1.0.

LDQOL 1.0	Self-Reported Disability Days *F* (*p*-Value)	Self-Rated Severity of Liver Disease Symptoms *F* (*p*-Value)	Child-Pugh Class *F* (*p*-Value)	Duration of Liver Disease *F* (*p*-Value)
Physical functioning	20.76 (<0.001)	26.70 (<0.001)	16.01 (<0.001)	2.39 (0.94)
Role limitations - physical	22.18 (<0.001)	29.24 (<0.001)	9.39 (<0.001)	1.69 (0.94)
Role limitations - emotional	17.21 (<0.001)	17.08 (<0.001)	12.54 (<0.001)	1.32 (0.94)
Social functioning	8.88 (<0.001)	16.50 (<0.001)	6.57 (<0.001)	1.05 (0.94)
Bodily pain	14.41 (<0.001)	28.72 (<0.001)	8.32 (<0.001)	0.60 (0.94)
Energy/fatigue	6.90 (<0.001)	14.43 (<0.001)	3.29 (0.027)	1.81 (0.94)
Emotional well-being	6.28 (<0.001)	6.05 (<0.001)	2.24 (0.12)	2.28 (0.94)
General health perceptions	10.39 (<0.001)	14.08 (<0.001)	2.04 (0.15)	1.51 (0.94)
Symptoms of liver disease	11.00 (<0.001)	14.06 (<0.001)	23.81 (<0.001)	2.31 (0.94)
Effects of liver disease	13.92 (<0.001)	24.08 (<0.001)	9.76 (<0.001)	1.73 (0.94)
Concentration	9.38 (<0.001)	28.18 (<0.001)	4.34 (0.006)	2.34 (0.94)
Memory	4.29 (0.004)	10.17 (<0.001)	1.09 (0.45)	2.06 (0.94)
Quality of social interaction	3.96 (0.006)	8.74 (<0.001)	1.58 (0.28)	3.82 (0.94)
Health distress	10.93 (<0.001)	24.03 (<0.001)	1.52 (0.29)	0.71 (0.94)
Sleep	5.78 (<0.001)	10.64 (<0.001)	3.61 (0.018)	1.01 (0.94)
Loneliness	5.47 (<0.001)	7.77 (<0.001)	5.70 (<0.001)	2.04 (0.94)
Hopelessness	4.72 (0.002)	11.00 (<0.001)	2.10 (0.10)	1.83 (0.94)
Stigma of liver disease	10.28 (<0.001)	11.27 (<0.001)	6.55 (<0.001)	1.85 (0.94)
Sexual functioning	4.68 (0.006)	10.89 (<0.001)	2.33 (0.13)	0.80 (0.94)
Sexual problems	8.64 (<0.001)	19.46 (<0.001)	5.13 (0.002)	1.19 (0.94)

LDQOL, liver disease quality of life instrument.

**Table 5 jcm-08-00656-t005:** Cox proportional hazards model of score changes of the LDQOL 1.0 components for death and decompensation.

Variables (Last score − first score, per 10 point)	Death		Decompensation	
	HR (95% CI)	*p*-Value	HR (95% CI)	*p*-Value
Physical functioning	0.98 (0.93‒1.04)	0.47	0.98 (0.94‒1.01)	0.17
Role limitations – physical	0.87 (0.46‒1.66)	0.67	1.02 (0.66‒1.60)	0.92
Role limitations ‒ emotional	1.21 (0.72‒2.03)	0.46	0.90 (0.61‒1.32)	0.58
Social functioning	1.06 (0.56‒1.99)	0.87	0.90 (0.60‒1.35)	0.61
Bodily pain	0.80 (0.52‒1.21)	0.28	0.90 (0.64‒1.26)	0.54
Energy/fatigue	0.91 (0.54‒1.49)	0.67	0.75 (0.54‒1.05)	0.09
Emotional well-being	0.73 (0.40‒1.34)	0.31	0.99 (0.95‒1.03)	0.48
General health perceptions	1.13 (0.60‒2.10)	0.71	0.86 (0.55‒1.32)	0.48
Symptoms of liver disease	0.40 (0.16‒1.02)	0.06	0.84 (0.35‒2.02)	0.70
Effects of liver disease	1.01 (0.92‒1.10)	0.86	0.70 (0.38‒1.31)	0.27
Concentration	0.44 (0.24‒0.78)	0.005	0.97 (0.94‒1.00)	0.029
Memory	0.58 (0.30‒1.15)	0.12	0.81 (0.49‒1.33)	0.40
Quality of social interaction	0.76 (0.44‒1.31)	0.32	0.83 (0.55‒1.25)	0.37
Health distress	0.59 (0.35‒1.00)	0.05	0.81 (0.63‒1.23)	0.31
Sleep	1.01 (0.52‒1.99)	0.97	1.04 (0.65‒1.68)	0.87
Loneliness	0.92 (0.52‒1.61)	0.76	0.74 (0.51‒1.05)	0.09
Hopelessness	0.69 (0.35‒1.33)	0.27	0.80 (0.55‒1.16)	0.23
Stigma of liver disease	0.95 (0.54‒1.66)	0.85	1.20 (0.85‒1.71)	0.30
Sexual functioning	0.98 (0.93‒1.03)	0.50	1.00 (0.95‒1.05)	0.84
Sexual problems	0.99 (0.92‒1.06)	0.75	0.98 (0.94‒1.02)	0.31

LDQOL, liver disease quality of life instrument; HR, hazard ratio; CI, confidence interval.

**Table 6 jcm-08-00656-t006:** Comparison of descriptive statistics according to the viral suppression status at the last visit.

LDQOL 1.0	Score at Last Visit			Score Change between the First and Last Visit		
	Complete Viral Suppression (n = 99)	No Viral Suppression (n = 92)	*p*-Value	Complete Viral Suppression (n = 78)	No Viral Suppression (n = 68)	*p*-Value
Physical functioning	89.22 ± 15.81	90.00 ± 17.26	0.78	+0.75 ± 17.11	−2.05 ± 16.27	0.33
Role limitations – physical	85.80 ± 24.28	87.78 ± 15.96	0.56	+5.99 ± 19.27	+0.37 ± 15.79	0.06
Role limitations – emotional	86.54 ± 21.67	87.62 ± 19.02	0.75	+3.88 ± 22.61	+2.24 ± 20.29	0.65
Social functioning	85.88 ± 21.87	87.50 ± 19.46	0.64	+2.36 ± 19.04	−1.14 ± 18.18	0.27
Bodily pain	88.65 ± 18.00	92.13 ± 15.14	0.21	+1.77 ± 19.67	+0.95 ± 18.24	0.80
Energy/fatigue	62.02 ± 22.09	62.78 ± 19.87	0.76	+2.33 ± 22.46	+1.33 ± 17.83	0.77
Emotional well-being	72.19 ± 19.69	70.82 ± 18.70	0.63	+1.27 ± 17.78	−0.38 ± 16.42	0.57
General health perceptions	59.11 ± 20.34	53.72 ± 19.22	0.07	+5.69 ± 20.18	−0.61 ± 14.61	0.037
Symptoms of liver disease	86.66 ± 14.10	87.90 ± 11.85	0.58	+1.08 ± 12.54	+0.04 ± 7.69	0.58
Effects of liver disease	83.70 ± 12.42	83.80 ± 12.00	0.96	+1.75 ± 13.13	−0.25 ± 13.92	0.40
Concentration	91.79 ± 15.00	91.38 ± 15.45	0.87	−0.74 ± 14.91	−2.16 ± 14.10	0.56
Memory	87.50 ± 15.29	89.26 ± 13.87	0.41	−5.88 ± 12.13	+0.57 ± 15.43	0.006
Quality of social interaction	74.90 ± 14.29	73.24 ± 16.86	0.47	−0.19 ± 16.03	−3.26 ± 13.66	0.23
Health distress	79.36 ± 22.58	79.88 ± 19.96	0.87	+8.49 ± 20.67	+3.69 ± 20.66	0.17
Sleep	65.27 ± 17.08	66.53 ± 19.33	0.69	+1.67 ± 17.84	+0.19 ± 17.65	0.64
Loneliness	78.47 ± 20.41	80.85 ± 16.88	0.46	−2.87 ± 18.85	−1.89 ± 16.71	0.76
Hopelessness	66.88 ± 21.86	69.44 ± 20.41	0.41	−0.16 ± 22.55	−1.61 ± 15.20	0.66
Stigma of liver disease	80.88 ± 23.13	82.84 ± 20.55	0.59	+2.40 ± 22.87	−3.06 ± 20.60	0.14
Sexual functioning	86.79 ± 17.97	90.86 ± 16.72	0.18	−2.64 ± 20.19	+0.88 ± 11.91	0.35
Sexual problems	80.02 ± 20.84	86.18 ± 14.22	0.07	+0.41 ± 18.30	+3.70 ± 16.75	0.39

LDQOL, liver disease quality of life instrument; SD, standard deviation.

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
