# Peer review of "Validation of the Liver Disease Quality of Life Instrument 1.0 in Patients with Chronic Hepatitis B: A Prospective Study"

_jcm, 2019, doi:10.3390/jcm8050656_

Reviewer 1 Report

 Comments to the Authors:

 The authors, Yeonjung Ha. et al., report usefulness of liver disease quality of life instrument (LDQOL) 1.0, which is prospectively measured in chronic hepatitis b patients.

They conclude that the decrease in concentration score predicted death and decompensation. Regular assessment of LDQOL 1.0 would be helpful and might be utilized further in developing clinical or policy guidelines.   

 The article is very interesing and can be useful in the clinical practise.

But the discussion should be revised one more time in my opinion.
Major comments:

1. Discussion is too long and redundant. The authors should summarize the key points of the clinical symptoms which decrease QOL in liver cirrhosis and chronic hepatitis B.

 Minor comments:

Is “SF 36 core” correct in your country?

Author Response

April 12th, 2019

 RE: JCM-478437.R1

TITLE: Validation of Liver Disease Quality of Life Instrument 1.0 in Patients with Chronic Hepatitis B: A Prospective Study

 We thank you and reviewers for their careful and thoughtful consideration of our work. We believe that these suggestions and our responses to them have improved the manuscript. Please find our point-by-point responses to the comments below.

 Comments by Reviewer 1
#1. Discussion is too long and redundant. The authors should summarize the key point of the clinical symptoms which decrease QOL in liver cirrhosis and chronic hepatitis B.

 Reply:

We appreciate and agree with the reviewer’s comment. The Results section also included some redundant information that is not directly related to the aim of our study. Such information might lead to a lengthy Discussion; therefore, we reviewed the whole manuscript very carefully, removed information of less clinical significance, and updated the manuscript accordingly. For example, we focused more on describing the importance of change in concentration score over time, which is a main finding of our study and is clinically relevant. An additional explanation/interpretation about other cognitive or mental function-related components was minimized in the Discussion section. In addition, general facts about questionnaire survey study (e.g., ceiling effect) were omitted.

All the changes are marked with the “track changes” function in the manuscript.

#2. Is “SF 36 core” correct in your country?

 Reply:

Thank you for this important comment. The Korean version of Short-Form 36 (SF 36) has been developed and validated in 2004 (Han CW et al. Tohoku J Exp Med. 2004 Jul;203(3):189-94). Although the study was validated on older adults with a mean age of 74 years, subsequent studies utilized SF 36 as a generic measure of health-related quality of life (HRQOL) in various populations such as general population with a mean age of  45 years (Kim SH et al. Asian Nurs Res (Korean Soc Nurs Sci). 2013 Jun;7(2):61-6), patients with irritable bowel syndrome and a mean age of 43 years (Park JM et al. Qual Life Res. 2009 May;18(4):435-46), and patients with Graves’ ophthalmopathy and a mean age of 41 years (Lee H et al. Korean J Ophthalmol. 2010 Apr;24(2):65-72). Therefore, we believe it is appropriate to use SF 36 as a tool measuring general health-related QOL in our patient population.

Reviewer 2 Report

The authors conducted the prospective study to validate the Liver Disease Quality of Life Instrument 1.0 for using practically among the patients with chronic hepatitis B in predicting their clinical outcomes especially overall survival and decompensation. 

 Major Comments

1.    It is valuable that study with negative result is reported. 

However, as for this manuscript, description of methodology including sampling method and analysis, and tables were too poor. For example, Table 4 is not able to understand without explanation of each value and p-value (it does not seem Cronbach’s alpha value). Result from Cox’s regression in Table 6 is too strange because all HRs are quite equal 1.00. It may come from issue of multicollinearity, and re-analysis should be considered. In Table 8, there is average score at last visit by liver disease state at baseline, but some of “No cirrhosis” changed to “Cirrhosis” during study period. It is also some of “Cirrhosis” died or loss of follow up. Therefore, it is difficult to evaluate the score in Table 8 so that the current study result does not lead the conclusion. 

2.    How did the authors define Complete Viral suppression in this study? How did the authors handle for those anti-viral naïve patients who did not take any kind of anti-viral treatment in subdividing complete viral suppression and non-suppression groups? 

3.    For subdividing the cirrhotic and non-cirrhotic groups, how about the non-cirrhotic patients at the baseline but decompensated to cirrhosis or HCC during follow-up visit? Because the number of cirrhotic patients at last visit (table 8) is the same as the number at the baseline (table 1). 

4.    The authors described the anti-viral naïve CHB group. It is curious is there any score changes between anti-viral naïve and anti-viral treated group and how about the validity of the LDQOL among anti-viral naïve group in clinical setting. 

 5.    The calculation for the table 4 is unclear? What are the values for each box and what the p-value represent for? What are the parameters used to define for self-rated severity of liver disease and Child-Pugh class (A or B or C)?

Minor comments

1.    Table 3 is difficult to understand for the readers because of putting the decimal numbers in two different rows. The landscapes style can be used. 

2.    In line no 113, the median HBV DNA level is 1.45x 105copies/ml but not 1.45x 105copies/ml.

 Author Response

April 12th, 2019

 RE: JCM-478437.R1

TITLE: Validation of Liver Disease Quality of Life Instrument 1.0 in Patients with Chronic Hepatitis B: A Prospective Study

 We thank you and reviewers for their careful and thoughtful consideration of our work. We believe that these suggestions and our responses to them have improved the manuscript. Please find our point-by-point responses to the comments below.

 Comments by Reviewer 2

 #1. It is valuable that study with negative result is reported. However, as for this manuscript, description of methodology including sampling method and analysis, and tables were too poor. For example, Table 4 is not able to understand without explanation of each value and p-value (it does not seem Cronbach’s alpha value). Result from Cox’s regression in Table 6 is too strange because all HRs are quite equal 1.00. It may come from issue of multicollinearity, and re-analysis should be considered. In Table 8, there is average score at last visit by liver disease state at baseline, but some of “No cirrhosis” changed to “Cirrhosis” during study period. It is also some of “Cirrhosis” died or loss of follow up. Therefore, it is difficult to evaluate the score in Table 8 so that the current study result does not lead the conclusion.

 Reply:

We are grateful to know that our current approach requires some rethinking.

First, as the reviewer commented, it will not be easy for readers to interpret the values shown in Table 4. The F value in the analysis of variance (ANOVA), which is used to demonstrate construct validity, is calculated from dividing ‘variance (variation) between groups’ by ‘variance (variation) within groups’ (F = variance bwetween groups / variance within groups). Therefore, by definition, larger F value indicates: 1) greater difference in average values between groups, and 2) lesser difference in average values within one group. In other words, if F value is high, we can say that the groups are different, but the difference is not due to some extreme values in one or more group. Whether the difference is significant or not is determined by the calculated P-value. As multiple comparisons are performed in the ANOVA, post-hoc correction was examined by using Benjamin-Hochberg method.

For example, in our analysis in Table 4, the ‘role limitations – physical’ component showed F value of 29.24 and P-value < 0.001 when assessed for the association with ‘self-rated severity of liver disease symptoms.’ It indicates that the average score of ‘role limitations – physical’ component was largely and significantly different according to the severity of liver disease symptoms (group 1: no symptoms; group 5: extremely severe symptoms). Supplementary Figure 1 below clearly shows that patients with the best (100) score had no (group 1) symptoms (blue arrow). Meanwhile, patients with the worst (0) score had moderate (group 3) symptoms (red arrow).

(Please refer to the attached file for Supplementary Fig. 1)

Supplementary Fig. 1. Boxplot showing the association between scores of the role limitation – physical component (x-axis) and self-rated severity of liver disease symptoms (y-axis).

 On the other hand, the association between ‘bodily pain’ score and the duration of liver disease was not significant (P = 0.94) with the F value of 0.60 (Supplementary Fig. 2).

(Please refer to the attached file for Supplementary Fig. 2)

Supplementary Fig. 2. Boxplot showing the association between the scores of bodily pain component (x-axis) and duration of liver disease (y-axis).

We modified the part describing statistical analysis for construct validity in the Method section as follows:  “For assessing construct validity, an analysis of variance (ANOVA) was performed to evaluate the association between each questionnaire component and the common parameters of liver disease. (…) The result of ANOVA was presented with the F values showing between group differences (F = variance between groups / variance within groups) and the P-values were adjusted by Benjamin-Hochberg method.” In addition, the result of the construct validity analysis was also updated with an easy-to-understand example in the Results section as follows: “The F values in Table 4 represent the differences in each component score according to the different severity of liver disease. All the component scores significantly differed according to the self-reported disability days and self-rated severity of liver disease symptoms. The ‘role limitation – physical component’ indicated large differences in scores between patients who rated their liver disease symptoms as more vs. less severe (F = 29.24; P < 0.001). The scores also significantly differed according to Child-Pugh class for 13 out of 20 components. On the other hand, all of the components showed comparable scores, regardless of the duration of patients’ liver disease (all P = 0.94).”

Second, we did the re-analysis for identifying questionnaire components that are associated with death or decompensation. As the reviewer pointed out, the hazard ratios (HRs) for each component were close to 1.00. It is likely due to the fact that the score of each component is on a 100-point scale and we calculated the hazard ratio of death or decompensation according to the increment of only 1 point. For example, as we stated in the Discussion section, HR 0.99 means the risk of death is decreased by 1%, when the score is increased (improved) by 1 point. However, the score actually changes by 10 or higher points unit; for instance, if the patient responded that his/her symptoms of liver disease changed from mild to moderate, the patient gets 20 points. Therefore, we re-calculated the HRs for death or decompensation with the same method that we had used but based on 10-point score increment for each component.

In addition, we included all the components, rather than the ones that showed statistical significance in the construct validity, because the component could be associated with death or decompensation even if it did not show an association with the common parameters of liver disease.

Moreover, after further discussion with the statistical expert (SHH), we did not define ‘time-varying covariate’ based on the paired sample t-test. Because we do not know if the score would be constant through the study period, it might not be appropriate to designate the time-varying covariate based on the result that we got later (Vermunt 1999 J Educ Behav Stat).

After the re-analysis, Table 6 is revised to Table 5 in the original manuscript and updated in the Results section as follows:

 Table 5 (revised from Table 6 in the original manuscript). Cox proportional hazards model of score change in LDQOL 1.0 component for death and decompensation

Variables

(Last score – first score, per 10 point)

Death

Decompensation

HR (95% CI)

P-Value

HR (95% CI)

P-Value

Physical functioning

0.98 (0.93‒1.04)

0.47

0.98 (0.94‒1.01)

0.17

Role limitations – physical

0.87 (0.46‒1.66)

0.67

1.02 (0.66‒1.60)

0.92

Role limitations ‒ emotional

1.21 (0.72‒2.03)

0.46

0.90 (0.61‒1.32)

0.58

Social functioning

1.06 (0.56‒1.99)

0.87

0.90 (0.60‒1.35)

0.61

Bodily pain

0.80 (0.52‒1.21)

0.28

0.90 (0.64‒1.26)

0.54

Energy/fatigue

0.91 (0.54‒1.49)

0.67

0.75 (0.54‒1.05)

0.09

Emotional well-being

0.73 (0.40‒1.34)

0.31

0.99 (0.95‒1.03)

0.48

General health perceptions

1.13 (0.60‒2.10)

0.71

0.86 (0.55‒1.32)

0.48

Symptoms of liver disease

0.40 (0.16‒1.02)

0.06

0.84 (0.35‒2.02)

0.70

Effects of liver disease

1.01 (0.92‒1.10)

0.86

0.70 (0.38‒1.31)

0.27

Concentration

0.44 (0.24‒0.78)

0.005

0.97 (0.94‒1.00)

0.029

Memory

0.58 (0.30‒1.15)

0.12

0.81 (0.49‒1.33)

0.40

Quality of social interaction

0.76 (0.44‒1.31)

0.32

0.83 (0.55‒1.25)

0.37

Health distress

0.59 (0.35‒1.00)

0.05

0.81 (0.63‒1.23)

0.31

Sleep

1.01 (0.52‒1.99)

0.97

1.04 (0.65‒1.68)

0.87

Loneliness

0.92 (0.52‒1.61)

0.76

0.74 (0.51‒1.05)

0.09

Hopelessness

0.69 (0.35‒1.33)

0.27

0.80 (0.55‒1.16)

0.23

Stigma of liver disease

0.95 (0.54‒1.66)

0.85

1.20 (0.85‒1.71)

0.30

Sexual functioning

0.98 (0.93‒1.03)

0.50

1.00 (0.95‒1.05)

0.84

Sexual problems

0.99 (0.92‒1.06)

0.75

0.98 (0.94‒1.02)

0.31

 Consistent with the original result, change in ‘concentration’ score was only associated with death or decompensation (marked as red). Multivariate analysis was not performed considering the fact that only one variable (concentration) was associated with the clinical outcomes and to avoid multicollinearity. Updated HRs are easier to interpret (56% decreased risk of death per 10 point increase in concentration score). The manuscript was updated according to the revised methodology and results (marked with ‘track changes’ function in the manuscript file).

Lastly, we agree with the reviewer that non-cirrhotic patients might have progressed to cirrhosis and some of cirrhotic patients could have been lost to follow-up due to death or clinical deterioration. In such cases, the score at last visit could erroneously be lower in ‘non-cirrhosis’ group and higher in ‘cirrhosis’ group. Accordingly, it would mask the score change that potentially existed between the two groups. Therefore, we decided to remove Table 8 from the manuscript. All the contents regarding the analysis according to baseline cirrhotic status were removed from the entire manuscript (marked with ‘track changes’ function in the manuscript file).

#2. How did the authors define Complete Viral suppression in this study? How did the authors handle for those anti-viral naïve patients who did not take any kind of anti-viral treatment in subdividing complete viral suppression and non-suppression groups?

 Reply:

We appreciate this comment.

First, complete viral suppression was defined as undetectable serum hepatitis B virus (HBV) DNA by real-time PCR assay when completing the last questionnaire. We added the definition of complete viral suppression in the Method section as follows: “The viral suppression status was evaluated by real-time PCR assay (Roche Diagnostics) when patients completed the last LDQOL 1.0 questionnaire and defined as undetectable hepatitis B virus (HBV) DNA in the serum.”

Second, we only used the above mentioned definition when subdividing patients into complete viral suppression vs. non-suppression group. The use of nucleos(t)ide analogs (NUCs) at study entry as well as over the study period did not affect the subgrouping of the patients. If HBV DNA was undetectable at the time of last LDQOL 1.0 completion, the patient was classified as ‘complete viral suppression’ group, regardless of the use of NUCs. As undetectable HBV DNA is associated with less inflammation (normal transaminase levels), less fibrosis, and thereby favorable prognosis (European Association for the Study of the Liver, J Hepatol. 2017 Aug;67(2):370-398), we classified the patients whose HBV DNA levels were undetectable, either naturally or medically, as ‘complete viral suppression’ group.

 #3. For subdividing the cirrhotic and non-cirrhotic groups, how about the non-cirrhotic patients at the baseline but decompensated to cirrhosis or HCC during follow-up visit? Because the number of cirrhotic patients at last visit (table 8) is the same as the number at the baseline (table 1).

 Reply:

We appreciate this comment. As the reviewer commented, 140 patients did not have cirrhosis and the remaining 52 patients had cirrhosis at baseline. Twenty-nine (20.7%) out of 140 non-cirrhotic patients and 16 (30.8%) out of 52 cirrhotic patients completed the LDQOL 1.0 questionnaire only once. For those 29 and 16 patients, the baseline score (their only score), was considered as ‘score at last visit’ in Table 8 (first visit = last visit). For calculating the score change between first and last visit, we included patients who completed the LDQOL 1.0 at least twice; therefore the number decreased to 111 and 36 for no cirrhosis and cirrhosis group, respectively.

Additionally, none of the patients developed hepatocellular carcinoma (HCC) during the study period. We do not have the information about how many non-cirrhotic patients progressed to cirrhosis, because collecting the patients’ cirrhotic status at last visit was not included in our study protocol, and thereby not was not approved by the institutional review board.

However, as we replied in comment #1 above, Table 8 was removed as the subgrouping of patients into no cirrhosis vs. cirrhosis according to baseline status might be masking the difference that potentially existed between the two groups.

 #4. The authors described the antiviral naïve CHB group. It is curious is there any score changes between anti-viral naïve and anti-viral treated group and how about the validity of the LDQOL among anti-viral naïve group in clinical setting.

 Reply:

Thank you very much for this helpful comment. Although we focused on the status of viral replication itself, either naturally or medically, we think it would be interesting to further analyze the change in the questionnaire score according to medication status. The Supplementary Table 1 below shows the results of our analysis.

 Supplementary Table 1. Comparison of descriptive statistics according to the NUCs treatment status at baseline.

LDQOL 1.0

Score   change between first   and last visit

No NUCs at baseline

(n = 28)

NUCs at baseline

(n = 120)

P-value

Physical   functioning

+8.80 ± 19.22

+1.22 ± 15.57

0.006

Role limitations   – physical

+2.00 ± 21.85

+3.50 ± 16.86

0.70

Role limitations   – emotional

-0.33 ± 22.88

+3.81 ± 21.09

0.38

Social   functioning

+0.96 ± 22.03

+0.54 ± 17.86

0.92

Bodily pain

+0.09 ± 14.57

+1.66 ± 19.76

0.70

Energy/fatigue

+2.40 ± 22.30

+1.78 ± 19.92

0.89

Emotional   well-being

-0.38 ± 18.70

-0.78 ± 16.77

0.75

General health   perceptions

+1.88 ± 19.10

+2.74 ± 17.75

0.83

Symptoms of   liver disease

-1.41 ± 10.96

+1.15 ± 10.32

0.28

Effects of liver   disease

-1.46 ± 17.73

+1.38 ± 12.23

0.45

Concentration

-1.40 ± 19.67

-1.38 ± 12.99

0.99

Memory

-3.57 ± 18.35

-2.71 ± 12.88

0.82

Quality of social interaction

-3.04 ± 19.16

-1.25 ± 13.83

0.65

Health distress

+9.60 ± 26.38

+5.45 ± 19.10

0.44

Sleep

-1.67 ± 18.76

+1.65 ± 17.37

0.39

Loneliness

-0.77 ± 20.96

-2.84 ± 16.93

0.60

Hopelessness

-0.23 ± 18.22

-1.08 ± 19.72

0.84

Stigma of liver   disease

-6.48 ± 26.94

+1.19 ± 20.59

0.10

Sexual   functioning

-0.98 ± 18.84

-1.19 ± 16.25

0.96

Sexual problems

+7.48 ± 21.64

+0.79 ± 16.34

0.17

NUC, nucleos(t)ide analogs

 As seen in the Table above, the extent of score change between the first and last visit was not statistically significant for all components, except for ‘physical functioning,’ between NUC-naïve vs. NUC-treated group at baseline. Additionally, the absolute extent of score change is small (-3.57 to +9.60), again prompting the regular assessment of QOL in CHB patients regardless of antiviral treatment.

Of note, some of the NUC-naïve patients at baseline might have started antiviral treatment during the study period, according to our clinical practice guideline (Korean Association for the Study of the Liver, Clin Mol Hepatol. 2016 Mar;22(1):18-75) and the reimbursement policy of National Health Insurance Service. That is, ‘no NUCs at baseline’ group in the Supplementary Table 1 may not have been completely antiviral-naïve, potentially resulting in a similar error that we had mentioned regarding Table 8 (removed) in the comment #1 above. Therefore, we would like to present Supplementary Table 1 here in the response letter but not in the original manuscript.

 #5. The calculation for the table 4 is unclear? What are the values for each box and what the p-value represent for? What are the parameters used to define for self-rated severity of liver disease and Child-Pugh class (A or B or C)?

 Reply:

We used ANOVA to show the association between each questionnaire component and common parameters of liver disease (self-rated severity of liver disease, Child-Pugh class, and so on). The numbers are F values (F statistics) derived from ANOVA, and the P-values show whether the association between the questionnaire component and parameters of liver disease is significant after multiple comparisons.

Per the reviewer’s comment here and in the comment #1, we stated the statistical methodology in detail in the Method section as follows: “For assessing construct validity, an analysis of variance (ANOVA) was performed to evaluate the association between each questionnaire component and the common parameters of liver disease. (…) The result of ANOVA was presented with the F values showing between group differences (F = variance between groups / variance within groups) and the P-values were adjusted by Benjamin-Hochberg method.” In addition, we modified the paragraph in the Results section as follows: “The F values in Table 4 represent the differences in each component score according to the different severity of liver disease. All the component scores significantly differed according to the self-reported disability days and self-rated severity of liver disease symptoms. The ‘role limitation – physical component’ indicated large differences in scores between patients who rated their liver disease symptoms as more vs. less severe (F = 29.24; P < 0.001). The score also significantly differed according to Child-Pugh class for 13 out of 20 components. On the other hand, all of the components showed comparable score, regardless of the duration of patients’ liver disease (all P = 0.94).” Finally, we added F in the heading of Table 4 for clear understanding.

Regarding parameters of liver disease, self-rated severity of liver disease was measured using a five-point categorical response scale (1 = no symptoms; 2 = mild symptoms; 3 = moderate symptoms; 4 = severe symptoms; and 5 = extremely severe symptoms). Child-Pugh class was categorized as A, B, and C according to Child-Pugh score calculated from albumin, bilirubin, prothrombin time, and the presence of ascites or hepatic encephalopathy. A detailed explanation about the common parameters of liver disease was added in the Method section as follows: “The common parameters of liver disease included 1) self-reported disability days (0 vs. 1-10 vs. 11-20 vs. ≥ 21 days in the preceding month), 2) self-rated severity of liver disease symptoms (from 1 [no symptoms] to 5 [extremely severe symptoms]), 3) Child-Pugh class (A vs. B vs. C), and 4) duration of liver disease (from 1 [less than 6 months] to 7 [more than 10 years] based on the diagnosis date on the medical record).”

 #6. Table 3 is difficult to understand for the readers because of putting the decimal numbers in two different rows. The landscapes style can be used.

 Reply:

We have adjusted the format of Table 3 for better readability.

 #7. In line no 113, the median HBV DNA level is 1.45 × 105 coplies/mL but not 1.45 × 105 copies/mL.

 Reply:

Thank you for your comment. We have modified this part accordingly.

Round  2

Reviewer 1 Report

A whole report is well written. 

Their viewpoint is highly original. 

But the Table 3. LDQOL 1.0 product-moment correlation should be revised for the reader can understand. 

Author Response

Done

Reviewer 2 Report

The authors conducted the prospective study to validate the Liver Disease Quality of Life Instrument 1.0 for using practically among the patients with chronic hepatitis B in predicting their clinical outcomes especially overall survival and decompensation. 

Major Comments

1.    It is valuable that study with negative result is reported. 

However, as for this manuscript, description of methodology including sampling method and analysis, and tables were too poor. For example, Table 4 is not able to understand without explanation of each value and p-value (it does not seem Cronbach’s alpha value). Result from Cox’s regression in Table 6 is too strange because all HRs are quite equal 1.00. It may come from issue of multicollinearity, and re-analysis should be considered. In Table 8, there is average score at last visit by liver disease state at baseline, but some of “No cirrhosis” changed to “Cirrhosis” during study period. It is also some of “Cirrhosis” died or loss of follow up. Therefore, it is difficult to evaluate the score in Table 8 so that the current study result does not lead the conclusion. 

2.    How did the authors define Complete Viral suppression in this study? How did the authors handle for those anti-viral naïve patients who did not take any kind of anti-viral treatment in subdividing complete viral suppression and non-suppression groups? 

3.    For subdividing the cirrhotic and non-cirrhotic groups, how about the non-cirrhotic patients at the baseline but decompensated to cirrhosis or HCC during follow-up visit? Because the number of cirrhotic patients at last visit (table 8) is the same as the number at the baseline (table 1). 

4.    The authors described the anti-viral naïve CHB group. It is curious is there any score changes between anti-viral naïve and anti-viral treated group and how about the validity of the LDQOL among anti-viral naïve group in clinical setting. 

 5.    The calculation for the table 4 is unclear? What are the values for each box and what the p-value represent for? What are the parameters used to define for self-rated severity of liver disease and Child-Pugh class (A or B or C)?

Minor comments

1.    Table 3 is difficult to understand for the readers because of putting the decimal numbers in two different rows. The landscapes style can be used. 

2.    In line no 113, the median HBV DNA level is 1.45x 105copies/ml but not 1.45x 105copies/ml.

Author Response

Done

Round  3

Reviewer 2 Report

I checked revised points and I agree to accept this manuscript to be published.

Author Response

Done